anxiety; depression; stress; male adults; perceived social support

**关键词:**
焦虑; 抑郁; 压力; 成年男性; 领悟社会支持

**Corresponding authors:**
Wei Li and Ruike Zhang;
Emails: 18994318368@189.cn; zrk_2015@163.com

T.H., X.M., and C.N. these authors contributed equally to the work.

‡This article has been updated since original publication. A notice detailing the change has been published.

# Prevalence and predictors of psychological distress among Chinese male adults: A cross-sectional study in a large general population‡

Tianya Hou[1] iD, Xiaofei Mao[1], Chunyan Ni[2], Jianguo Zhang[1], Wenxi Deng[1], Wei Li[3] and Ruike Zhang[1] iD

[1]Faculty of Psychology, Naval Medical University, China; [2]Suzhou Hospital, Affiliated Hospital of Medical School, Nanjing University, China and [3]Nanjing University Medical School Affiliated Nanjing Drum Tower Hospital Department, China

## Abstract

Since little is known about the mental health status of Chinese male adults, the present study aimed to explore the prevalence and predictors of psychological distress among Chinese male adults. This cross-sectional study recruited 6,140 men by convenience sampling in Liaoning and Shanghai. Psychological distress and perceived social support were measured by the Depression, Anxiety, and Stress Scale-21 and the Perceived Social Support Scale. Multivariable logistic regression analysis was employed to investigate the factors associated with psychological distress. The prevalence of depression, anxiety and stress among Chinese male adults were 14.2%, 26.4% and 12.9%, respectively. Male adults with low perceived social support reported symptoms on all measurements. Being middle-aged, college or above and residing in urban areas were associated with depression, while living in urban areas was the independent risk factor of anxiety. Stress was significantly associated with being middle-aged, married status and college or above. Our results revealed a high prevalence of psychological distress among Chinese male adults in this sample. Results suggest more attention needs to be paid to the mental health of male adults in the studied population, especially those who are middle-aged, college or above, married status and reside in urban areas.

## 摘要

目前关于中国成年男性心理健康状况的研究较为匮乏。本研究旨在探讨中国成年男性心理困扰的流行率及其预测因素。这项横断面研究通过便利抽样法在辽宁和上海招募了6，140名男性，采用抑郁-焦虑-压力量表(DASS-21)和领悟社会支持量表(PSSS)评估其心理困扰程度和社会支持感知水平，并运用多因素logistic回归分析相关影响因素。结果显示: 中国成年男性的抑郁、焦虑和压力症状检出率分别为14.2%、26.4%和12.9%。社会支持感知水平较低的男性在所有测量指标上均表现出症状。中年、大专及以上学历、城市居住与抑郁显著相关，而城市居住是焦虑的独立危险因素。压力症状则与中年、已婚状态及大专及以上学历显著相关。本研究结果显示，样本中的中国男性成年人心理困扰普遍率较高,提示应重点关注中年、高学历、已婚及城市居住男性群体的心理健康状况。

## Impact statements

This study provides critical insights into the mental health challenges faced by Chinese male adults, a population that has been understudied in psychological research. By analyzing data from 6,140 participants, we reveal high prevalence rates of depression (14.2%), anxiety (26.4%) and stress (12.9%), highlighting a significant public health concern. Our findings identify key risk factors, including middle age, higher education (college or above), urban residence and marital status, with low perceived social support emerging as a consistent predictor across all psychological distress measures. The results underscore the urgent need for targeted mental health interventions and policies tailored to Chinese male adults, particularly those in urban settings and with higher education levels. By raising awareness of these vulnerable groups, our research can inform healthcare providers, policymakers and community organizations in developing culturally appropriate support systems, such as workplace mental health programs and accessible counseling services. This study not only fills a gap in the existing literature but also has broader implications for improving mental health outcomes in similar demographic groups globally, where traditional gender norms may discourage help-seeking behaviors among men. Ultimately, our work advocates for a more inclusive approach to mental health care that addresses the unique needs of adult men in rapidly urbanizing societies.

## Background

China has undergone unprecedented economic growth and social changes since the reform and opening up, leading to enormous alterations in agriculture, industry, education, disease epidemiology and so on (Huang et al., 2019). Rapid social changes are likely to result in a general enhancement of psychological distress – a broad term encompassing negative emotional states such as depression, anxiety and stress (Huang et al., 2019). According to the China Mental Health Survey (CMHS), the weighted lifetime incidence rate of any mental illness (excluding dementia) was 16.6%, with anxiety being the most common mental disorder in China (Huang et al., 2019). Depressive disorders have been the second leading cause of disability in China (Gao et al., 2023; Chen et al., 2025).

Anxiety refers to a subjective feeling of persistent, intense and excessive fear, while depression is viewed as a state of low mood accompanied by a loss of interest in daily activities. Stress refers to the experience of both physical and emotional tension caused by any life events that threaten the homeostasis (Rehman et al., 2021). Taken together, these conditions contribute significantly to the overall burden of psychological distress in the population. Given their high prevalence and functional impact, mental health issues – especially depression, anxiety and stress – have become both a clinical and public health priority in contemporary China.

Several factors have been demonstrated to be associated with the occurrence of mental health problems, including sociodemographic and psychological factors. Evidence from different literature has suggested several sociodemographic factors associated with mental health conditions, such as age, gender, marital status, educational level, sibling status and children status and place of residence (Cao et al., 2020; Hou et al., 2020; Lu et al., 2021; Basta et al., 2022; Liu et al., 2022; Zhang et al., 2023; Humer et al., 2025; Jónsdóttir et al., 2025). In addition, existing literature has investigated the impact of social support as a psychological factor on mental health (Hou et al., 2020; Rhubart and Kowalkowski, 2022; Juarez et al., 2025; Ross and Ross, 2025). Social support refers to the support and care individuals perceive or experience from others (Raschke, 1978). The buffer hypothesis proposed by Cohen and Wills (1985) suggested that social support could help buffer and protect individuals from the detrimental effects of stressful events, contributing to mental well-being. Although the majority of the studies presented a positive association between social support and mental health status (Hou et al., 2020), there were still a few studies reporting the opposite findings in some contexts (Nicolini et al., 2021).

Several studies have investigated the incidence rate and influential factors of mental problems (León Rojas et al., 2022; Nahar et al., 2022; Tyagi and Meena, 2022; Brečka et al., 2024). The majority of the findings have shown that psychological distress was more prevalent in women than men (Lu et al., 2021; Carter et al., 2025; Khorasani et al., 2025), resulting in a substantial number of studies on the mental health status of female adults (Jeon et al., 2020; Marano et al., 2025; Pasciuto et al., 2025). By contrast, the research focusing on male adults is less than females. Thus, the mental health conditions among male adults are a relatively underexplored research area. To fill the literature gap, the present study was conducted to investigate the prevalence and associated factors of psychological distress among male adults in China.

## Methods

### *Participants*

This cross-sectional study was conducted from November 2021 to January 2022. Data were collected through the online survey platform "Questionnaire Star" (Wenjuanxing) using both convenience and snowball sampling techniques. The survey link was distributed through various social platforms and was primarily disseminated within Liaoning Province and Shanghai Municipality.

As recommended by prior literature, we adopted the rule of thumb of 10 events per variable to ensure adequate statistical power (Wang et al., 2023). Given that our regression model included seven independent variables, a minimum sample size of 70 was determined. Respondents who met the following inclusion criteria were invited to participate in the research: (1) were aged ≥ 18 years old, (2) were males, (3) had no cognitive impairment and (4) provided written informed consent. A total of 6,185 males were eligible to participate in the study. The participants were excluded if they did not respond to any question or reported a history of mental illnesses. Finally, 6,140 questionnaires remained in the formal analysis.

The study was approved by the Medical Ethics Committee of Suzhou Science and Technology Town Hospital (IRB201912002RI). Written informed consent was obtained before the administration of the questionnaire. Subjects were informed that their participation was voluntary and were assured that all responses were anonymous and confidential.

### *Measures*

#### *Demographic characteristics*
Demographic information was collected, including age (≤30 vs. >30), marital status (unmarried vs. married), only child (yes vs. no), children situation (no child vs. one child or more), educational level (high school or below vs. college or above) and place of residence (rural vs. urban).

#### *Perceived social support*
The Chinese version of the perceived social support scale was used to assess the perception of social support (Jiang, 2001). The 12-item scale measures the social support from family, friends and significant others. On a 7-point Likert scale (ranging from 1 "*very strongly disagree*" to 7 "*very strongly agree*"), respondents were asked to indicate how they feel about each item. The total score was calculated by adding the scores of all 12 items and ranged from 12 to 84, with scores 12–36, 37–60 and 61–84 representing low, fair and high perceived social support. The scale has been widely used among the Chinese population with good validity and reliability (Zhang et al., 2022). In the present study, the Cronbach's alpha was 0.940.

#### *Depression, anxiety and stress scale*
The 21-item Depression, Anxiety and Stress Scale (DASS-21) was employed to assess depression, anxiety and stress. Participants were required to rate each item on a 4-point Likert scale ranging from 0 (*never*) to 3 (*almost always*). The subscale score was calculated by summing up the item values and multiplying by 2. According to the DASS manual, the subscale score was further categorized into normal, mild, moderate, severe and extremely severe.

The total score of the depression subscale ranged from 0 to 42 and could be classified as normal (0–9), mild (10–13), moderate (14–20), severe (21–27) and extremely severe (≥28) depression. The total score of the anxiety subscale ranged from 0 to 42 and could be regarded as normal (0–7), mild (8–9), moderate (10–14), severe (15–19) and extremely severe (≥20) anxiety. The total score of the stress subscale ranged from 0 to 42 and could be divided as normal (0–14), mild (15–18), moderate (19–25), severe (26–33) and extremely severe (≥34) stress.

The scale has been demonstrated to have good validity and reliability in Chinese adults (Guo et al., 2022). The Cronbach's alpha values were 0.873, 0.817 and 0.850 for the depression, anxiety and stress subscales, respectively. Considering mild to extremely severe as reporting depression, anxiety and stress, the cutoff points for detecting symptoms of major depression, anxiety and stress were 10, 8 and 15, respectively.

### Statistical analysis

Statistical Package for Social Sciences (SPSS v.21.0; SPSS Inc., Chicago, IL, USA) was employed for data analysis. Categorical variables were presented as frequencies and percentages. A binary logistic regression model with 95% confidence interval (CI) was performed to screen the independent predictors of binary outcome variables (Natnael et al., 2021). We conducted both bivariate (crude odds ratio [COR]) and multivariate (adjusted odds ratio [AOR]) logistic regression analyses. First, each factor was separately included in the bivariate analysis, and a COR with 95% CI was adopted to investigate the crude association between each factor and mental health outcomes (Tamir et al., 2021). Second, all factors were entered into the multivariate analysis for further analysis regardless of the significance level of predictors in bivariate analysis (Njiro et al., 2021). An AOR with a 95% CI was utilized to identify the influential factors associated with mental health outcomes. Stratified analyses across all demographic subgroups were performed using logistic regression to estimate CORs and 95% CIs for associations between perceived social support levels and outcomes of depression, anxiety and stress.

### Results

#### Demographic characteristics of the respondents

Table 1 showed that the majority of the respondents were aged 30 years or under (79.2%). More than half, 3,638 (59.3%), of the respondents were unmarried. About two-thirds of the respondents were not the only child (63.3%) and did not have any children (68.7%). About three-quarters of the participants reported an educational level of college or above (72.1%) and were residents in rural areas (73.2%). In terms of perceived social support, most participants reported high perceived social support (78.6%).

#### Prevalence of depression, anxiety and stress

The overall prevalence of depression, anxiety and stress in this study was found to be 14.2%, 26.4% and 12.9%, respectively. As shown in Figure 1, 462 (7.5%) subjects reported mild depression, 320 (5.2%) subjects reported moderate depression, 56 (0.9%) subjects reported severe depression and 33 (0.5%) reported extremely severe depression. There were 7.3%, 14.4%, 3.0% and 1.6% respondents who had mild, moderate, severe and extremely severe anxiety,

**Table 1.** Demographic characteristic of male adults

| Variables | Categories | Frequency | Percent |
|---|---|---|---|
| Age | | | |
| | Young group (18 ~ 30 years) | 4,473 | 72.9 |
| | Middle-aged group (31 ~ 54 years) | 1,667 | 27.1 |
| Marital status | | | |
| | Unmarried | 3,638 | 59.3 |
| | Married | 2,502 | 40.7 |
| Only child | | | |
| | Yes | 2,255 | 36.7 |
| | No | 3,885 | 63.3 |
| Children situation | | | |
| | No child | 4,220 | 68.7 |
| | One child or more | 1920 | 31.3 |
| Educational level | | | |
| | High school or below | 1711 | 27.9 |
| | College or above | 4,429 | 72.1 |
| Place of residence | | | |
| | Rural | 4,492 | 73.2 |
| | Urban | 1,648 | 26.8 |
| Perceived social support | | | |
| | Low-to-fair | 1,312 | 21.4 |
| | High | 4,828 | 78.6 |

respectively. In addition, the incident rates of mild, moderate, severe and extremely severe stress were 7.6%, 3.7%, 1.4% and 0.3%, respectively.

#### Prevalence and predictors of depression among male adults

Table 2 presented bivariate and multivariate analyses of factors associated with depression among male adults. In bivariate analysis, age, marital status, children status, educational level, place of residence and perceived social support were factors significantly associated with depression. Following that, multiple logistic regression was conducted to identify the individual contributions of each factor when controlling for other factors. The results of multivariate analyses showed that age, educational level, place of residence and perceived social support were still significantly associated with depression.

Young subjects were found to have lower odds of having depression (AOR = 0.613, 95% CI = 0.472–0.796) than middle-aged subjects. The odds of male adults developing depression were found to be lower for participants who reported an educational level of high school or below (AOR = 0.775, 95% CI = 0.641–0.937) than those who reported an educational level of college or above. Participants who lived in rural areas were at a decreased risk of having depression (AOR = 0.749, 95% CI = 0.634–0.886) than those who lived in urban areas. Compared with participants with high perceived social support, participants with low-to-fair perceived social support had 6.891 times the likelihood (AOR = 6.891, 95% CI = 5.899–8.051) to report depression.

(a)

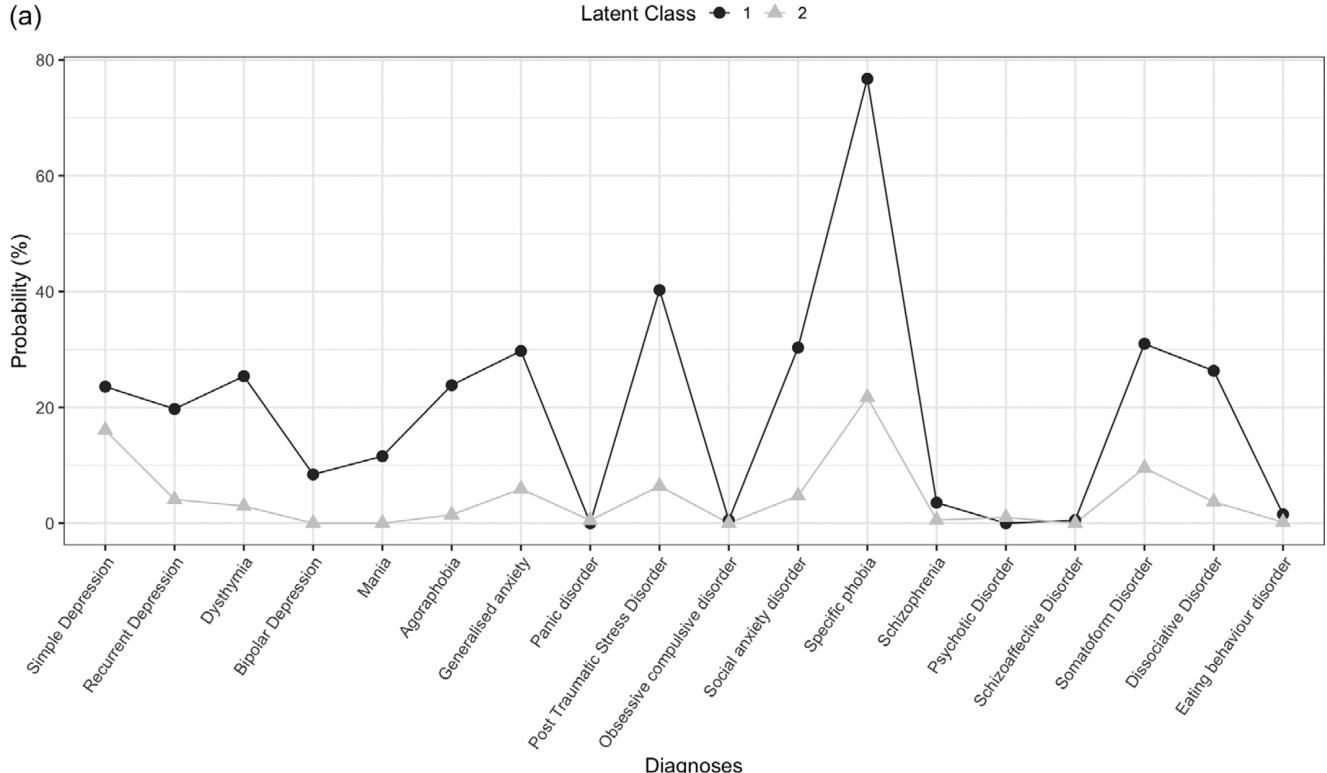

(b)

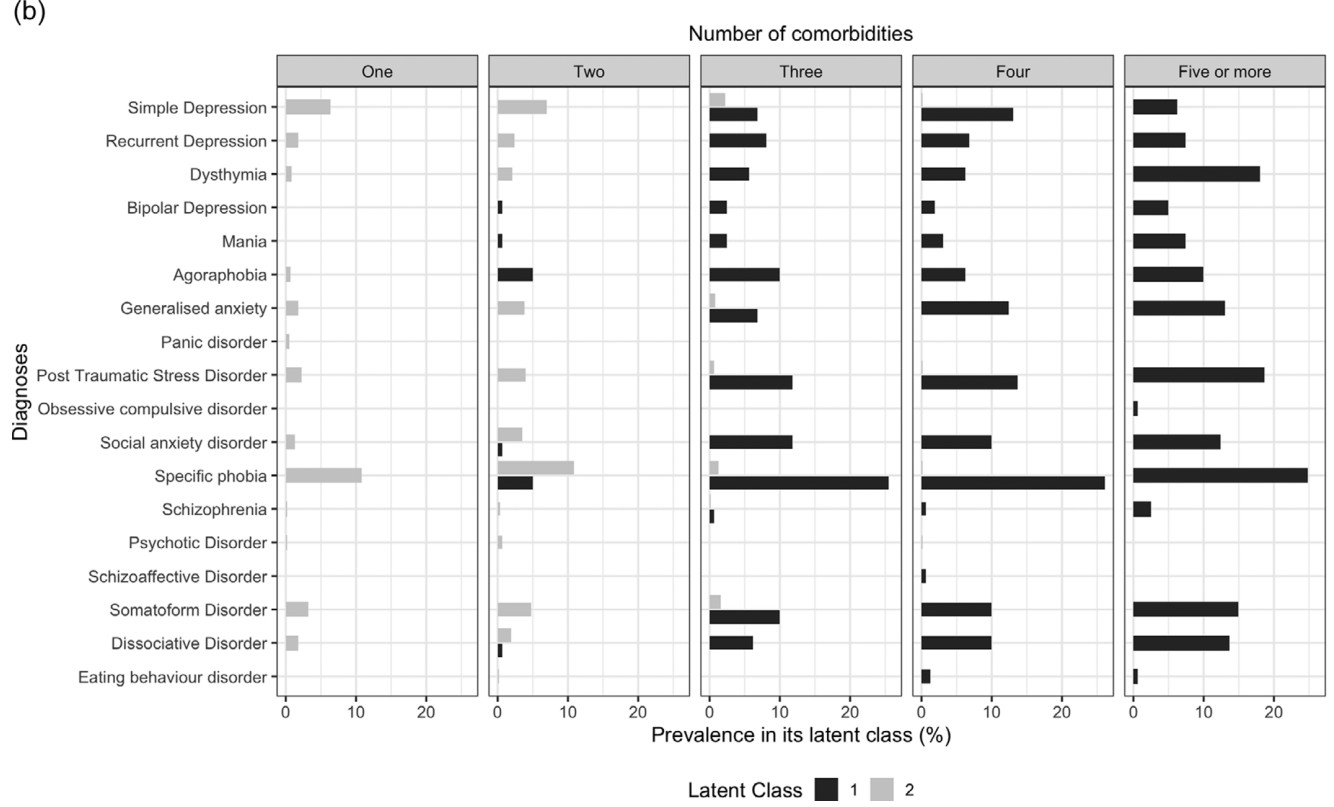

**Figure 1.** Severity of depression, anxiety and stress for male respondents in China (*n* = 6,140).

**Table 2.** Bivariate and multivariate analyses of factors associated with depression among male adults

| Variables | Category | Depression | | COR and 95% CI | AOR and 95% CI | p-value |
|---|---|---|---|---|---|---|
| | | No | Yes | | | |
| Age | | | | | | |
| | Young group | 3,920 (87.6) | 553 (12.4) | 0.598 (0.514–0.696) | 0.613 (0.472–0.796) | <0.001 |
| | Middle-aged group | 1,349 (80.9) | 318 (19.1) | 1 | 1 | |
| Marital status | | | | | | |
| | Unmarried | 3,190 (87.7) | 448 (12.3) | 0.69 (0.598–0.797) | 0.935 (0.711–1.229) | 0.631 |
| | Married | 2,079 (83.1) | 423 (16.9) | 1 | 1 | |
| Only child | | | | | | |
| | Yes | 1,945 (86.3) | 310 (13.7) | 0.944 (0.813–1.097) | 1.104 (0.938–1.299) | 0.233 |
| | No | 3,324 (85.6) | 561 (14.4) | 1 | 1 | |
| Children situation | | | | | | |
| | No child | 3,688 (87.4) | 532 (12.6) | 0.673 (0.58–0.78) | 0.976 (0.715–1.332) | 0.879 |
| | One child or more | 1,581 (82.3) | 339 (17.7) | 1 | 1 | |
| Educational level | | | | | | |
| | High school or below | 1,509 (88.2) | 202 (11.8) | 0.752 (0.636–0.89) | 0.775 (0.641–0.937) | 0.009 |
| | College or above | 3,760 (84.9) | 669 (15.1) | 1 | 1 | |
| Place of residence | | | | | | |
| | Rural | 3,906 (87) | 586 (13) | 0.717 (0.615–0.837) | 0.749 (0.634–0.886) | 0.001 |
| | Urban | 1,363 (82.7) | 285 (17.3) | 1 | 1 | |
| Perceived social support | | | | | | |
| | Low-to-fair | 828 (63.1) | 484 (36.9) | 6.708 (5.757–7.816) | 6.891 (5.899–8.051) | <0.001 |
| | High | 4,441 (92) | 387 (8) | 1 | 1 | |

### Prevalence and predictors of anxiety among male adults

The bivariate and multivariate analyses of factors associated with anxiety among male adults were shown in Table 3. In the bivariate logistic regression analysis, anxiety was significantly associated with age, marital status, children situation, place of residence and perceived social support. In the multivariate analysis, place of residence and perceived social support remained significant. Participants residing in rural areas were less likely to have anxiety (AOR = 0.771, 95% CI = 0.676–0.880) than those residing in urban areas. Low-to-fair perceived social support (AOR = 4.275, 95% CI = 3.752–4.870) was a significant predictor for anxiety.

### Prevalence and predictors of stress among male adults

The bivariate and multivariate analyses of factors associated with stress among male adults were presented in Table 4. The results of bivariate analysis showed that age, marital status, children situation, educational level, place of residence and perceived social support were significantly associated with stress among male adults. Age, marital status, educational level and perceived social support were the factors significantly associated with stress in the final model. Young male adults were at a decreased risk of developing stress (AOR = 0.658, 95% CI = 0.509–0.851) than middle-aged adults. Being unmarried was a protective factor for stress (AOR = 0.766, 95% CI = 0.588–0.998). The odds of having stress were significantly lower for males with an educational level of high school or below (AOR = 0.820, 95% CI = 0.677–0.994) compared to those with a level of college or above. In comparison to males with high social support, the likelihood of suffering from stress was 3.803 times more prevalent in those with low-to-fair social support (AOR = 3.803, 95% CI = 3.245–4.457).

### Associations of perceived social support with depression, anxiety and stress stratified by demographic variables

Stratified analyses across all demographic subgroups (see Supplementary Table S1) – including age, marital status, only-child status, parenthood, education level and place of residence – consistently demonstrated significantly elevated CORs for depression, anxiety and stress among individuals with low-to-fair perceived social support compared to the high support reference group (all COR > 3.3, 95% CIs excluding 1). Although the strength of association varied slightly between subgroups, with unmarried individuals and those without siblings exhibiting higher point estimates (e.g., COR for depression = 7.662 and 8.041, respectively), the adverse associations remained statistically significant and directionally consistent in every stratum. These results underscore the robust and pervasive protective role of high perceived social support against poor mental health outcomes, independent of diverse demographic backgrounds.

### Discussion

The present study, which included 6,140 respondents, revealed a high prevalence of mental health symptoms among Chinese male

**Table 3.** Bivariate and multivariate analyses of factors associated with anxiety among male adults

| Variables | Category | Anxiety | | COR and 95% CI | AOR and 95% CI | *p*-value |
|---|---|---|---|---|---|---|
| | | No | Yes | | | |
| Age | | | | | | |
| | Young group | 3,360 (75.1) | 1,113 (24.9) | 0.756 (0.667–0.856) | 0.851 (0.694–1.045) | 0.125 |
| | Middle-aged group | 1,159 (69.5) | 508 (30.5) | 1 | 1 | |
| Marital status | | | | | | |
| | Unmarried | 2,745 (75.5) | 893 (24.5) | 0.793 (0.707–0.889) | 0.942 (0.762–1.164) | 0.579 |
| | Married | 1,774 (70.9) | 728 (29.1) | 1 | 1 | |
| Only child | | | | | | |
| | Yes | 1,679 (74.5) | 576 (25.5) | 0.932 (0.828–1.049) | 1.03 (0.909–1.169) | 0.640 |
| | No | 2,840 (73.1) | 1,045 (26.9) | 1 | 1 | |
| Children situation | | | | | | |
| | No child | 3,175 (75.2) | 1,045 (24.8) | 0.768 (0.681–0.866) | 0.865 (0.677–1.106) | 0.247 |
| | One child or more | 1,344 (70) | 576 (30) | 1 | 1 | |
| Educational level | | | | | | |
| | High school or below | 1,276 (74.6) | 435 (25.4) | 0.932 (0.821–1.059) | 0.956 (0.829–1.102) | 0.532 |
| | College or above | 3,243 (73.2) | 1,186 (26.8) | 1 | 1 | |
| Place of residence | | | | | | |
| | Rural | 3,379 (75.2) | 1,113 (24.8) | 0.739 (0.653–0.837) | 0.771 (0.676–0.880) | <0.001 |
| | Urban | 1,140 (69.2) | 508 (30.8) | 1 | 1 | |
| Perceived social support | | | | | | |
| | Low-to-fair | 641 (48.9) | 671 (51.1) | 4.273 (3.754–4.864) | 4.275 (3.752–4.870) | <0.001 |
| | High | 3,878(80.3) | 950 (19.7) | 1 | 1 | |

adults. Specifically, 14.2%, 26.4% and 12.9% of participants reported symptoms of depression, anxiety and stress, respectively. Low perceived social support was significantly associated with all three types of symptoms.

While certain demographic factors – such as middle age, higher education (college or above), urban residence and being married – were also associated with higher reports of specific symptoms, the most consistent and strong correlate across all mental health outcomes was low social support. These findings highlight concerning patterns in the psychological well-being of male adults in China and suggest that low social support may play an important role in mental health outcomes in this population.

The study found that the overall prevalence of depression among Chinese male adults was 14.2%, which was notably higher than that reported in previous literature (Lu et al., 2021; Cheng et al., 2022). Evidence from nationally representative data revealed the weighted lifetime and 12-month prevalence of depression among male adults in China was 5.7% and 3.0%, respectively (Lu et al., 2021). Another recent study explored the overall prevalence of depressive disorders among the adult population in Shandong province of China and reported the incidence rate of depressive disorders in men was 3.48% (Cheng et al., 2022). Notably, this elevated prevalence aligns with global patterns where depressive disorders show no significant difference between low- and high-income countries, with a global prevalence of 3.8% (including 2.49% for major depressive disorder and 1.35% for

dysthymia) (Castaldelli-Maia and Bhugra, 2022). Moreover, since 2010, China has experienced a more rapid increase in depression Disability-Adjusted Life Years compared to the global average, and all national income classifications are consistent with a growing mental health burden (Chen et al., 2024).

Similarly, our results indicated that more than one in four male adults in our sample reported symptoms of anxiety, which is higher than the weighted lifetime and 12-month prevalence of 7.6% and 5.0% among Chinese adults reported in a recent study based on the CMHS (Huang et al., 2019). This discrepancy may be attributed to several factors, including the specific focus on males in our study – a group that may underreport mental health issues in general population surveys due to stigma or gender-related social expectations.

In the present study, about one in seven male adults experienced stress in China, a rate that appears higher than the prevalence of stress (8.8%) observed among Chinese university students (Cheung et al., 2020). The broader age range and diverse socioeconomic backgrounds of our sample may capture a wider spectrum of stress exposures among adult men in the general population, extending beyond the academic-specific pressures commonly faced by student populations. These comparisons suggest that Chinese male adults may face unique and pervasive psychosocial challenges that are associated with higher levels of stress and anxiety.

Together, these findings highlight an urgent need to develop gender-sensitive mental health screening protocols and support mechanisms tailored specifically to adult men. The elevated rates

**Table 4.** Bivariate and multivariate analyses of factors associated with stress among male adults

| Variables | Category | No | Yes | COR and 95% CI | AOR and 95% CI | *p*-value |
|---|---|---|---|---|---|---|
| Age | | | | | | |
| | Young group | 3,976 (88.9) | 497 (11.1) | 0.579 (0.495–0.677) | 0.658 (0.509–0.851) | 0.001 |
| | Middle-aged group | 1,371 (82.2) | 296 (17.8) | 1 | 1 | |
| Marital status | | | | | | |
| | Unmarried | 3,248 (89.3) | 390 (10.7) | 0.625 (0.538–0.726) | 0.766 (0.588–0.998) | 0.048 |
| | Married | 2,099 (83.9) | 403 (16.1) | 1 | 1 | |
| Only child | | | | | | |
| | Yes | 1,964 (87.1) | 291 (12.9) | 0.999 (0.855–1.166) | 1.142 (0.971–1.344) | 0.109 |
| | No | 3,383 (87.1) | 502 (12.9) | 1 | 1 | |
| Children situation | | | | | | |
| | No child | 3,744 (88.7) | 476 (11.3) | 0.643 (0.551–0.75) | 1.076 (0.796–1.454) | 0.634 |
| | One child or more | 1,603 (83.5) | 317 (16.5) | 1 | 1 | |
| Educational level | | | | | | |
| | High school or below | 1,531 (89.5) | 180 (10.5) | 0.732 (0.614–0.873) | 0.820 (0.677–0.994) | 0.043 |
| | College or above | 3,816 (86.2) | 613 (13.8) | 1 | 1 | |
| Place of residence | | | | | | |
| | Rural | 3,937 (87.6) | 555 (12.4) | 0.835 (0.709–0.984) | 0.857 (0.722–1.016) | 0.075 |
| | Urban | 1,410 (85.6) | 238 (14.4) | 1 | 1 | |
| Perceived social support | | | | | | |
| | Low-to-fair | 957 (72.9) | 355 (27.1) | 3.718 (3.179–4.348) | 3.803 (3.245–4.457) | <0.001 |
| | High | 4,390 (90.9) | 438 (9.1) | 1 | 1 | |

of anxiety and stress symptoms are consistent with the possibility that current public health approaches may not be fully addressing the mental health burdens reported by this demographic.

In accordance with the results from the previous literature, middle-aged male adults in our study reported higher levels of depression and stress than young male adults (Hou et al., 2021). This pattern is consistent with the distinct socioeconomic and familial responsibilities that characterize midlife. Middle-aged men often occupy a pivotal role within the family structure, frequently facing intensified occupational pressures as they strive to advance their careers or maintain job stability in a competitive environment. Simultaneously, they are typically responsible for providing financial and emotional support for both their children and aging parents, creating a "sandwich generation" effect that is associated with chronic stress and emotional exhaustion.

The present study revealed that obtaining higher educational degrees (college or above) was associated with depression and stress, which is in line with the findings from recent research (Belo et al., 2020; Zhang et al., 2023; Yan et al., 2025). Highly educated male adults may be more likely to work in a higher position and experience greater responsibilities, which co-occur with higher levels of psychological distress. It should be noted, however, that the association between education and stress was of marginal statistical significance, and the CI approached the null value. This suggests the result should be interpreted cautiously, as the effect may be sensitive to unmeasured confounders or sample-specific characteristics.

The present study further suggested that married male adults were more likely to experience stress. This result contrasts with previous findings that reported marriage was generally associated with protective effects against stress (Lindström and Rosvall, 2012; Walsh et al., 2023). In some Asian contexts, married men are often expected to assume the role of primary financial provider and carry greater familial responsibilities compared to unmarried men (James-Hawkins et al., 2019), which may coincide with higher levels of stress. Notably, the observed association was marginally significant, with the upper bound of the CI being very close to 1.00, indicating that the finding is statistically tentative. Further research with larger samples is warranted to confirm the direction and magnitude of this relationship.

The results showed living in urban areas was associated with depression and anxiety, which is in line with the findings from previous literature (Cao et al., 2020). Urban living has been repeatedly associated with higher rates of mental disorders. Urban residents may encounter a variety of challenges and stressors due to factors such as a fast pace of life, fierce job competition, high living costs and traffic congestion. These conditions often co-occur with higher reports of mental health symptoms among male adults in urban settings.

The association between lower perceived social support and poorer mental health outcomes was consistently observed in both unadjusted and demographically stratified analyses, further corroborating the robustness of this relationship and its consistency with prior epidemiological evidence (Hou et al., 2020; Bedaso

et al., 2021; Hou et al., 2021). Consistent with these studies, male adults who reported low levels of perceived social support were more likely to experience symptoms of depression, anxiety and stress. This pattern of results is consistent with the classic stress-buffering hypothesis proposed by Cohen and Wills (1985), which suggests that social support may act as a protective resource by mitigating the adverse effects of stressful experiences on psychological health.

Specifically, in the context of male adults, adequate social support – whether from family, friends or community networks – may provide crucial emotional reassurance, practical assistance and a sense of belonging, all of which are associated with enhanced capacity to cope with life challenges. Those with higher perceived social support tend to report greater access to both tangible and emotional resources that correlate with resilience in the face of economic pressure, familial responsibilities or work-related stress (Liu et al., 2024; Yang et al., 2024). In contrast, the absence of such support is often linked to increased reported sensitivity to stressors and fewer perceived avenues for relief or constructive problem-solving. Correspondingly, males with stronger social networks are less likely to report severe or persistent psychological symptoms, highlighting the potential importance of relational health in public mental health strategies, especially among populations traditionally reluctant to seek formal psychological help.

## Implications

Based on our results, early identification and monitoring systems for male adults with factors associated with higher risk of mental health conditions could be considered. Three levels of evidence-based prevention (universal interventions, selective interventions and indicated interventions) (Arango et al., 2018) may offer a structured approach. Universal interventions could focus on the promotion of the mental well-being of the general male population (e.g., community-based men's health promotion programs) (Oliffe et al., 2020). Selective interventions might target subgroups with specific risk factors (e.g., interventions for married male adults). The indicated interventions target the male adults who have already suffered from mental health conditions (e.g., social support programs for depressed male adults).

These strategies may help address a growing public health concern within China and could provide a scalable model for other middle-income countries facing similar mental health transitions. By integrating digital health technologies and existing primary care infrastructures, China may have an opportunity to develop cost-effective, population-level mental health interventions for often overlooked male populations.

## Strength and limitation

To the best of the researchers' knowledge, the present study is the first to explore the prevalence and associated factors of psychological distress among Chinese male adults with a large sample size. This represents a significant strength, as the extensive sample size enhances the statistical power and generalizability of the findings, providing a more reliable basis for understanding mental health challenges in this population.

Despite this strength, several limitations should be noted when interpreting the findings in the present study. First, the present study adopted a cross-sectional design and lacked longitudinal follow-up, which limits the inference of a causal relationship. Further study could use a longitudinal design to verify the association

between social support and psychological distress. Secondly, the data in the present study were obtained through self-reports. The results might be influenced by recall biases (Song and Sun, 2023). Further investigations could be carried out to collect information from multi-informants. Third, this study has limitations in sample representativeness. The use of online convenience and snowball sampling, restricted to Liaoning Province and Shanghai Municipality, resulted in a sample characterized by an overrepresentation of young, highly educated and rural participants. This demographic profile differs substantially from the broader Chinese male adult population in terms of age, educational background and geographical distribution. Although stratified analyses were conducted to mitigate potential bias, the generalizability of the results remains limited to populations with similar sociodemographic and regional characteristics. Future studies employing probabilistic sampling methods across more diverse geographical regions are warranted to enhance the external validity of the findings. Finally, psychological distress as a complex concept could be affected by numerous factors, including demographic, psychological and environmental factors. The influential factors in the present study could only account for a part of the variance. Further study could collect more information for a more comprehensive understanding of psychological stress among Chinese male adults.

## Conclusions

In conclusion, our results suggested that symptoms of depression, anxiety and stress were quite prevalent in the general male population of China. Special attention should be paid to high-risk male adults with the following characteristics: middle-aged, college degree or above, married status, residence in urban areas and low perceived social support.

**Open peer review.** To view the open peer review materials for this article, please visit http://doi.org/10.1017/gmh.2025.10114.

**Supplementary material.** The supplementary material for this article can be found at http://doi.org/10.1017/gmh.2025.10114.

**Data availability statement.** Data are available from the corresponding author based on reasonable requests.

**Acknowledgements.** The authors would like to thank all participants and cooperating authors in this study.

**Author contribution.** T.H.: Conceptualization, formal analysis, funding acquisition, writing – original draft. C.N., J.Z., W.D. and X.M.: Data curation, formal analysis. W.L.: Methodology, writing – review and editing. R.Z.: Funding acquisition, methodology, writing – review and editing. All authors read and approved the final manuscript for publication and take full responsibility for the work, including investigating and resolving any questions related to its accuracy or integrity.

**Financial support.** The authors disclosed receipt of the following financial support for the research, authorship and/or publication of this article: Shanghai Municipal Health Commission's Special Clinical Research Project for the Hygiene Industry (T.H., grant number 20244Y0041), Youth Initiation Fund of Second Military Medical University (T.H., grant number 2023QN028) and Youth Initiation Fund of Second Military Medical University (R.Z., grant number 2023QN030).

**Competing interests.** The authors declare none.

**Ethics approval.** All methods in the present study were carried out in accordance with relevant guidelines and regulations. The study involving human participants was reviewed and approved by the Medical Ethics Committee of

Suzhou Science and Technology Town Hospital (IRB201912002RI). The participants provided their written informed consent to participate in this study.

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
