## [Reviewer Report]

This study addresses a critical gap in global mental health literature by investigating understudied psychological distress in Chinese male adults. The large sample size (n=6,140) strengthens epidemiological validity. While the topic aligns well with the journal’s scope, several methodological and interpretive issues require revision before publication.

Major

1.The title claims a prospective study, yet methods describe a cross-sectional online survey (Nov 2021-Jan 2022).

2.The study’s failure to explicitly detail sampling frameworks (e.g., province-specific recruitment strategies, offline inclusion mechanisms) and demographic validation methods severely compromises its capacity to generalize findings to Chinese male adults. Without addressing the overrepresentation of young, highly educated, and rural participants - in stark contrast to China’s urban-majority demographics - the claimed predictors (e.g., urban residence risk) and prevalence rates lack national relevance.

3.Sample representation limitations.

3.1 Age bias: 72.9% participants 30 years (Table 1), limiting “middle-aged” conclusions (only 27.1% aged 31-54). Addictional stratified analysis is necessary.

3.2 Urban-rural imbalance: 73.2% rural vs. 26.8% urban, yet urban residence is flagged as risk factor.

We suggested addictional stratified analysis and discussion about the sampling limitations.

4.Statistical anomalies and interpretation.

4.1Implausible ORs: Low social support→depression (AOR = 119.8, 95% CI = 15.469 - 928.19, Table 2) suggests sparse data bias (only 0.2% in low-support group).

4.2Marginally significance: College-or-above education associated with stress (AOR = 0.823, p = 0.047, Table 4) with 95% CI of 0.679 - 0.998.

We suggested addictional discussion and cautious interpretation in results.

5.Causal overinterpretation in Discussion.

5.1 Highly-educated male adults are more likely to... bear greater responsibilities, causing higher levels of psychological distress

5.2 Married men are usually expected to earn money... which might explain why married men were more susceptible to stress.

5.3 ...

Minor

6.The specific relationship between psychological distress in title and depression, anxiety, and stress in the context should be explicitly elucidated in Introduction section.

7.The image tag references a low-resolution.

8.Update to include more recent studies (2023-2025).

---

## [Reviewer Report]

The perspective of this article focuses on the study of the mental health status of adult males, which deserves attention and further exploration, but the article still needs to make the following changes.

1. Please adjust the format of some parts of the article according to the requirements of the journal.

2. References should be richer and less literature before the last five years. Check the format of references.

3. Check the punctuation throughout the article.

4. Add the calculation of minimum sample size in the methodology section of the article.

5. The conclusion section should be listed separately.

6. Prevalence of depression in other countries could be added to the discussion section.

7.The discussion section should be simplified when it comes to demographic characteristics; the discussion of the prevalence of depression, anxiety and stress should be enriched; and the impact of social support on depression, anxiety and stress should be expanded, as there is too little space now.

8. Strength and Limitation need to be separated.

9. Implications could be increased to include more about the impact of the article on China or the world.

10. All Tables should be beautified and checked for details.

---

## [Reviewer Report]

It can be seen that the author has put in a great deal of effort to revise this article. I hope you will continue to strive and produce high-quality papers.

---

## [Reviewer Report]

1.The title case is incorrect and requires revision.

2.The Methods section of the abstract must explicitly describe this as a cross-sectional study, specify the sampling approach, and note the regional scope of participant recruitment. Conclusions should be qualified with phrases such as “in this sample” to avoid overgeneralization. Additionally, correct the grammatical error by changing “needs be paid” to “needs to be paid”.

3.The classification for “extremely severe stress” in the Methods section is incorrect. The cut-off score (≥24) must be verified and aligned with the standard DASS-21 scoring protocol.

4.In the Results section, ensure consistency between text and tables: the subsection on anxiety incorrectly references Table 3. Furthermore, the description of the association between education level and stress is misleading.

5.The submitted figure is of insufficient resolution for publication.